# An Analysis on the Determining Factors of Farmers' Land-Scale Management: Empirical Analysis Based on the Micro-Perspective of Farmers in China

**Xiang Li [1] and Hyukku Lee [2,*]**

[1] School of Business, Heze University, Heze 274015, China; lixiang@hezeu.edu.cn
[2] Department of East-Asia Studies, Graduate School, Pai Chai University, Daejeon 35345, Korea
* Correspondence: leehk@pcu.ac.kr; Tel.: +82-10-5567-0146

**Abstract:** In the context of continuous improvement in China's land system, the development of the rural economy is insufficient, and the growth of farmers' income lacks sustainable momentum. The development of the internet and agricultural socialization services has had a huge impact on farmers' land-scale management. In particular, the proliferation of internet technology in rural areas could affect farmers' use of agricultural socialization services and increase farmers' willingness to operate their land on a large scale. However, there is a lack of empirical evidence on the impact of the internet and agricultural socialization services on farmers' land-scale management decisions. This study constructs a probit model using the cross-sectional data of the nationally representative CFPS2018, and empirically tests the influencing factors of farmers' land-scale management decisions and the mechanism of heterogeneity. The research results show that, first, the popularity of the internet significantly promoted farmers' decisions towards land rented-out, but has no significant impact on land rented-in; second, agricultural socialization services are significantly negatively correlated with farmers' decisions towards land rented-out, but the internet may moderate this inhibitory effect and has an incentive effect on farmers' land rented-in; third, the results of heterogeneity analysis show that the impact of the internet and agricultural socialization services on farmers' land-scale management decisions vary with income levels and regions. Therefore, the policy direction should focus on making "internet + agriculture" and agricultural socialization services benefit all farmers, to more effectively improve the efficiency of rural land use and promote the optimal allocation of rural resources.

**Keywords:** land-scale management; internet; agricultural socialization services; probit model; heterogeneity; rural land; farmer; China

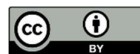

## 1. Introduction

Economic theories that explain the nature of institutions greatly contribute to understanding how various types of institutions affect resource allocation and innovation processes. As economic performance changes as institutions change, it is important to examine what changes institutions and what determines their direction. In particular, the various reforms being carried out in China are a process of institutional change. For a long time, "big country and small farmers" has been the basic pattern of Chinese agriculture. China's rural areas are facing the actual phenomenon of land fragmentation. Moderate-land scale management may be an important way to improve land-use efficiency [1]. Land transfer can transfer land from low-productivity households to high-productivity households, or from farmers who cannot cultivate land to those who need land for agricultural activities. Land circulation can improve the efficiency of resource allocation [2]. However, when describing the social picture of traditional Chinese farmers, that land is the "lifeblood" of farmers, and farmers are "stuck to the land" and lack mobility [3].

In the 1980s, China's rural market-oriented reform established a system of "separation of two rights" between land ownership and contracted management rights centered on the household contract system, and farmers gradually gained the freedom to leave the land [4]. In 2014, the system of "separation of three rights" was formally established to uphold the collective ownership of land, guarantee farmers' land contract rights and invigorate land management rights, encourage farmers to participate in the farmland leasing market, improve production efficiency, and use agricultural machinery and other labor-saving technologies [5]. In 2018, China promulgated and revised the Rural Land Contract Law, which aims to improve the efficiency of land use based on collective ownership of rural land. In 2021, the Ministry of Agriculture and Rural Affairs promulgated the "Administrative Measures for the Circulation of Rural Land Management Rights", which proposed that the circulation of land management rights should be adapted to local conditions step by step, and in line with improvement in the level of agricultural socialization services.

From the perspective of the transformation process of China's rural land property rights policy after the reform and opening up, rural land property rights, as an institutional tool of society, make it possible to shape the expectations that people may have in dealing with others [6], developed in the direction of farmers' expected economic benefits. The factor of institutional change lies in the expectation of economic agents to maximize potential profits. In other words, if it is difficult to expect more economic profits under the existing institutional structure, and institutional changes should be sought to obtain external profits [7]. Although China's rural land leasing system has a greater incentive, it forces farmers to take all risks, and policies to support land-scale management have not made China's land transfer develop rapidly [8]. The key factor is that land, as a special production factor, has serious principal–agent problems in the process of land circulation. The development of rural land reform in China over the past decade has confirmed that path dependence may lead to economic inefficiency, but this does not prove that the entire system is inefficient under the relevant economic constraints [9].

However, the cost of institutional change can be reduced with advances in science, technology, and knowledge, as well as the growth of social science knowledge [10], and institutional change can be achieved by reducing costs [7]. As technology has developed, the degree of information asymmetry between principals and agents has decreased [11]. In the modern agricultural production system, agricultural socialization services and internet technology have become the keys to affecting agricultural production. If so, has the change in China's rural land property rights system solved the problem of information asymmetry and reduced costs through the introduction of new technologies? With the popularization of internet technology in rural China, can farmers obtain enough information for decision making on land-scale management and improve the efficiency of resource allocation?

According to the existing literature, the research on agricultural socialization services and land-scale management is relatively detailed, but the research on linking them with internet technology is not sufficient. Based on this, the goal of this paper is to explore the impact of the internet and agricultural socialization services on farmers' decision making on land-scale management. Specifically, the statistical analysis of this paper is based on the data of the China Family Panel Studies (CFPS) of Peking University in 2018, and the probit model is used to explore the impact of the internet and agricultural socialization services on farmers' land-scale management decisions. Additionally, we analyze the moderating effect of the internet and the heterogeneity of farmers' decisions. The marginal contributions of this paper are reflected in: first, the paper pays attention to the influence of the internet and agricultural socialization services on farmers' land-scale management decisions from the theoretical and empirical levels; second, unlike previous studies, this study also focuses on the moderating effect of the internet. Third, this study explores the potential heterogeneous effects of different regions and income levels on land-scale management decisions, which can provide a more accurate refer-

ence for decision making. This paper is organized as follows. In Section 2, we introduce theoretical analysis and propose research hypotheses. Section 3 describes the empirical strategy and data selection. Section 4 is about empirical results, heterogeneity analysis, and robustness testing. Section 5 concludes.

## 2. Theoretical Analysis and Research Hypotheses

This paper mainly starts from the perspective of farmers' land-scale management, takes the internet and agricultural socialization services as the core, and explores the current decision making of farmers' land circulation. The reasons for choosing these three points as the core of the analysis are as follows. First, under the current institutional framework of "separation of three rights" in rural land in China, the government encourages large-scale operations through land transfer, thereby improving the production efficiency of agricultural land. The main body of decision making in land transfer is farmers, and their behavioral logic determines whether improvement in land efficiency can be realized. Second, agricultural socialization services refer to the support services provided by social and economic organizations or individuals for pre-, mid-, and post-production links in agriculture [1]. However, considering the reality of China, agricultural socialization services are more important to the in-production behavior of agricultural production. Therefore, the investigation of this paper mainly focuses on agricultural machinery leasing and labor services in agricultural production. Third, the development of the internet can enable farmers to obtain various information related to agricultural services through the internet. To a certain extent, this overcomes time constraints, reduces transaction costs, and has a positive effect on the optimal allocation of resources [12].

### 2.1. Internet and Land-Scale Management

With the emergence of large-scale farm operators such as family farms, cooperatives, and agricultural enterprises, China's rural land lease transactions have begun to shift from relational to market-oriented [13]. As of 2019, 23% of the farmland contracted by farmers' households was transferred to professional cooperatives, an increase of 4% over the previous year, and 12% was transferred to agricultural enterprises, an increase of 3.67% over the previous year. The area of exchanged land between farmers decreased by 10.39% over the previous year, but the proportion was still more than 50%, indicating that the scale and intensification level of land has been significantly improved. However, the core of land circulation is still the transaction between farmers, and the market-oriented land rental market has not formed. With the development of the marketization of land leasing in China, the internet as a medium can make market information more abundant and allocate resources more effectively. For example, the Tuliuwang APP provides farmers with a large amount of land circulation information. With improvement in rural internet infrastructure, more and more farmers have access to information through the internet, and the internet plays an increasingly important role in the process of large-scale land management.

The development of agricultural socialization services and the internet has further promoted social division of labor, leading to the initial emergence of large-scale farm operators in China [14]. When the agricultural production capacity of individual farmers is relatively high, they tend to rent farmland to achieve large-scale operations, so that agricultural socialization services and the internet can more fully play the role of reducing costs. At the same time, the emergence of large-scale farm operators can improve the agricultural production capacity of villages and even regions through demonstration effects [15]. The weakness of agriculture lies in the low added value of its products, the lack of specialized division of labor in agricultural production, and the short industrial chain of agricultural products. Some scholars have pointed out that it is necessary to develop modern agriculture with the idea of large-scale industry and rely on the expansion of the division of labor to lengthen the value chain of agricultural products and increase the added value of agricultural products [16]. However, for ordinary farmers, the cost of obtaining

agricultural socialization services and using the internet is relatively high, so leasing farmland may be a better choice. The existing literature pays more attention to the impact of agricultural socialization services or agricultural mechanization on agricultural production, and rarely takes the internet as a key research object. Some studies have pointed out that the high cost of mechanized services seems to force ordinary farmers to abandon mechanized services [17], but the penetration of the internet in rural areas may change this phenomenon.

Efficient land use has long been recognized as an important way to help increase land productivity and lift farmers out of poverty. Since 2004, agricultural socialization services and the internet have developed rapidly in rural China, and have formed an effective substitute for the lost labor force, enabling rural households with labor shortages to transfer land to expand their business scale [18]. Several studies have shown that these services significantly increase agricultural productivity [19]. In China, farmers constitute the vast majority of low-income groups. Agricultural socialization services can effectively control and reduce the huge sunk costs caused by farmers' self-purchasing of agricultural machinery [20]. The internet can effectively reduce farmers' transaction costs when choosing agricultural socialization services, thereby eliminating poverty [9]. In the "No. 1 central document" in 2019, the Chinese government took the development of agricultural socialization services and the acceleration of the modernization of ordinary farmers as important measures to consolidate and improve the basic rural management system. Some studies summarized this approach as "promoting agricultural modernization through service scale", and argued that realizing service scale in single or multiple links of agricultural production can not only enhance the profitability of service entities but also enable agricultural operators to reduce production costs and improve agricultural production efficiency, and form a win–win situation for production entities and service entities [21]. It is an important direction for China's agricultural transformation and agricultural modernization.

According to relevant research on the modern network economy, the use of the internet can bring information resources to farmers, greatly reduce information costs, and reduce information asymmetry [22]. As far as the transfer of farmland is concerned, first of all, the internet can reduce the cost for farmers to obtain information [23], especially in rural areas where the market is fragmented, and the internet can help farmers to participate in the market more effectively and transfer the scope of land transfer from acquaintances to switches between strangers. Second, farmers can obtain more agricultural information through the internet and change their behavioral decisions [24]. Finally, the internet promotes a fairer rural labor market, enables farmers to have more off-farm employment opportunities, and facilitates labor transfer [25]. Therefore, for the land rented-out party, the development of the internet has a strong promoting effect. However, for the land rented-in party, the purpose of land transfer is to achieve low-cost and large-scale operation. Then, the necessary condition must be that the land is contiguous. From the perspective of space, adjacent land in the same village or adjacent village is the optimal choice for the land rented-in [26]. Therefore, the land rented-in decision of farmers is more deeply influenced by acquaintances, and the influence of the internet is minimal. Many studies have shown that farmers often rely on the circulation information provided by their relatives, friends, and acquaintances [27]. Farmers with rich social networks in rural areas are often land rented-in parties whose credit endorsements are accepted by ordinary farmers [28,29], thereby renting land at lower transaction costs than the internet.

Therefore, we propose:

**Hypothesis 1**: The internet will increase the probability of farmers' decision to rent-out land, and the impact on farmers' decision to rent-in land is not clear.

## 2.2. Agricultural Socialization Service and Land-Scale Management

After China started the road of urbanization, a large number of farmers flooded into cities, resulting in a rapid increase in the opportunity cost of agricultural production. Agricultural socialization services play an important role in reducing agricultural production costs and promoting agricultural modernization. In rural China, the transfer of labor from agriculture to other sectors is increasing, and the advancement of agricultural production technology enables the labor-intensive processes in agricultural production to be replaced by machines, which will promote the socialization of agriculture service development [30]. The "No. 1 central document" in 2007 clearly stated for the first time that we should encourage the use of machinery, develop specialized agricultural services, and promote the marketization of agricultural socialization services. Since then, the machine-farming rate, machine-seeding rate, and yield of grain crops in China have all increased significantly, which has further promoted the development of agricultural socialization services [17].

Some scholars argue that, with the increase in farmers' income and the shortage of rural labor, there will be more and more need for agricultural socialization services, that is, outsourcing some agricultural production steps to professional service providers to reduce costs [31,32]. The empirical analysis of the village-level data shows that unified agricultural socialization services have a significant role in promoting land-scale management [33]. At the same time, the expansion of the operation scale of farmers will accelerate the dissemination of agricultural technology and agricultural machinery, and the expansion of the moderate operation scale has a positive effect on the outsourcing of agricultural production [30,34,35], but land fragmentation also leads to a high cost of agricultural socialization services [17]. The vigorous rise of the internet in rural areas has brought new opportunities for the development of agricultural socialization services.

Theoretically, suppose there is a rational farmer who behaves according to the principle of profit maximization. If the opportunity cost of agricultural production becomes high, he will rent-out the land and withdraw from agricultural production. Due to the migration of a large number of urban and rural populations, non-agricultural employment has become the main source of income for Chinese households. In large-scale operations, the widespread use of agricultural machinery can further reduce agricultural production costs and improve land production efficiency [36]. Since the sunk cost of purchasing agricultural machinery for individuals is too high, this highlights the huge cost advantage of agricultural socialization services. Therefore, we can expect that the availability of agricultural socialization services will reduce the production cost of farmers and improve the production efficiency, thereby reducing the land rented-out and increasing the land rented-in.

Therefore, we propose:

**Hypothesis 2:** Agricultural socialization services will reduce the probability of farmers' land rented-out and increase the probability of farmers' land rented-in.

## 2.3. Heterogeneous Perspective

Under the assumption of profit maximization, if the non-farm employment income is higher than the net profit of agricultural production, farmers are likely to increase the probability of renting out land. For now, the high opportunity cost and low income of agricultural production in China have prompted a large number of farmers to withdraw from agricultural production [17]. However, the large-scale operation of land requires the continuity of the plot. When the land is operated on a large scale, the land on the edge of the plot may be abandoned. This is due to the cost of large-scale operations. Therefore, the impact of agricultural socialization services on farmers' decision making is asymmetric, which means that in the process of agricultural modernization and the marketization of production factor allocation, some ordinary farmers will not be able to fully enjoy the low-cost advantage of agricultural socialization services. Similarly, the high opportunity cost of agricultural production leads to a shortage of rural labor. In addition, the high sunk

cost and high learning cost of self-owned agricultural machinery are not suitable for the production needs of ordinary farmers [37]. Farmers need to expand new channels to obtain various agricultural information, reducing transaction costs for agricultural production. For the land transfer party, the use of the internet can reduce market friction and improve the bargaining power of farmers [22], thereby effectively reducing the cost of farmland transfer and increasing the income of the transferring party. However, due to the different rural labor structures and terrain in different regions, the internet and agricultural socialization services will have a heterogeneous impact on the decision making of land-scale management.

According to the survey for six consecutive years from 2014 to 2019, more than 50% of market entities such as cooperatives and agricultural enterprises participating in the large-scale land transfer are dominated by returning hometown entrepreneurs [38]. The agricultural enterprises founded by the returning hometown entrepreneurs often participate in the land transfer process using "company + cooperatives (family farms) + farmers (large farming households) + bases", while returning hometown entrepreneurs with less capital from cooperatives or family farms join leading enterprises, and the land they transfer and operate has become the order production base for leading enterprises. For internet and agricultural socialization services, the use costs for them are lower than those of lower-income farmers. Therefore, for farmers with different income levels, the internet and agricultural socialization services have a heterogeneous impact on land-scale management decisions.

We propose:

**Hypothesis 3:** Differences in regions and income levels will have a heterogeneous impact on farmers' decision making on land-scale management.

## 3. Data, Research Model, and Variable Selection

### 3.1. Data

The data used in this study come from the China Family Panel Studies (CFPS) conducted by Peking University in 2018. The survey aims to collect data at three levels: individual, family, and community through tracking to reflect the demographic characteristics, income, and expenditure of Chinese families, agricultural production, economic activity, and non-economic welfare. The CFPS2018 national baseline survey covers 31 provinces (excluding Hong Kong, Macau, Taiwan, Xinjiang Uygur Autonomous Region, Tibet Autonomous Region, Qinghai Province, Inner Mongolia Autonomous Region, Ningxia Hui Autonomous Region, and Hainan Province), using a three-stage unequal probability cluster sampling design. Therefore, the data of CFPS2018 can be regarded as a nationally representative sample with good representation [39]. This article first merges the family data and personal data in the CFPS database. Secondly, due to the lack of village information in CFPS2018 and CFPS2016, this paper matches and merges the data of CFPS2014 with the data of CFPS2016 and CFPS2018, and retains the sample of rural members who participated in three surveys at the same time. Again, exclusion applied to 178 samples of personal information loss due to changes in family members, since CFPS2016 and CFPS2014 did not include Xinjiang Uygur Autonomous Region, Tibet Autonomous Region, Qinghai Province, Inner Mongolia Autonomous Region, Ningxia Hui Autonomous Region, and Hainan Province during the survey, data for these six provinces were removed when the data were merged. Finally, in the sorted CFPS2018 dataset, all rural residents were retained and the duplicated samples were eliminated. After review and sorting, a total of 4982 valid samples were obtained from 292 villages in 25 provinces.

### 3.2. Research Model

Theoretical analysis shows that the internet and agricultural socialization services have an impact on farmers' decision making on land-scale management. The explained variable of this study is a binary choice variable. Among the commonly used binary choice models, the probit model and the logit model are two more common models. The data source of this paper, CFPS, has a large sample, and the random interference term can be asymptotically approximated to a standard normal distribution by default [5,36,39–42], so the probit model is more suitable. The probit model in this paper is constructed as follows:

$$Landin_i = \alpha_0 + \beta_1 Inter_i + \beta_2 service_i + \beta_3 Inter_i * service_i$$
$$+ \sum_{j=0}^{7} \alpha_{Pj} P_{ji} + \sum_{j=0}^{4} \alpha_{Hj} H_{ji} + \sum_{j=0}^{2} \alpha_{Vj} V_{ji} + \varepsilon_1 \tag{1}$$

$$Landout_i = \alpha_0 + \beta_1 Inter_i + \beta_2 service_i + \beta_3 Inter_i * service_i$$
$$+ \sum_{j=0}^{7} \alpha_{Pj} P_{ji} + \sum_{j=0}^{4} \alpha_{Hj} H_{ji} + \sum_{j=0}^{2} \alpha_{Vj} V_{ji} + \varepsilon_2 \tag{2}$$

Equation (1) is the decision equation for farmers' land inflow; Equation (2) is the decision equation for farmers' land outflow. In the two formulas, the subscript $i$ represents the $i$th farmer, the subscript $j$ represents the $j$th variable, $Landin_i$ and $Landout_i$ represent the land-scale management decision variables, which are represented by whether the farmer rented-in or rented-out land; $service_i$ represents the agricultural socialization service decision variable, which is represented by whether the farmer rents agricultural machinery or hired labor; $Inter_i$ represents the internet decision variable, which is represented by whether the farmer uses the internet; $Inter_i*service_i$ is the interaction term, researching the moderating effect of the internet on agricultural socialization services. $P_{ji}$, $H_{ji}$, and $V_{ji}$ represent the individual characteristics, family characteristics, and village characteristics of farmers, respectively; $\varepsilon_1$, $\varepsilon_2$ are random error terms.

### 3.3. Variable Selection

The explanatory variable is a decision-making variable representing the scale of land management and is represented by whether the farmer rented-in or rented-out land, both of which are binary discrete variables. The existing literature is used for reference, and we use dummy variables to represent land-scale management decisions [43,44]. The core explanatory variables are agricultural socialization services and internet information acquisition, both of which are represented by dummy variables commonly used in the literature [12,17].

We also control for individual farmer, household, and village characteristics in our analysis. Farmers themselves are the most important decision makers in the family, so five control variables, including gender, age, education level, health status, and marital status, are introduced [45]. Some studies have pointed out that household characteristics determine the use of cultivated land [17]. According to existing research, family population size, the value of self-owned agricultural machinery, social relations, self-management, and per capita household net income are selected as proxy variables for family characteristics [12,36,46–48]. Similarly, non-agricultural employment may lead to land-waste and land-scale management [49], characterized by non-agricultural work income and the logarithm of all currency-related variables. The characteristics of villages are characterized by the distance from the village to the county seat and the topography of the village. These characteristics not only affect the development of comprehensive agricultural land but also determine the use of farmland by farmers [17,50]. Detailed definitions and statistical descriptions of dependent and independent variables are shown in Table 1.

**Table 1.** Descriptive statistics.

| Variable Name | Variable Description | Mean | Standard Deviation |
|---|---|---|---|
| landout | Land outflow: yes = 1; no = 0 | 0.151 | 0.358 |
| landin | Land inflow: yes = 1; no = 0 | 0.123 | 0.329 |
| internet | Whether uses the internet: yes = 1; no | 0.378 | 0.485 |
| service | Whether uses agricultural socialization services: yes = 1; no = 0 | 0.542 | 0.498 |
| sex | Farmer's gender | 0.514 | 0.500 |
| age | Farmer's age | 48.16 | 18.46 |
| edu | The educational level of farmers (years) | 2.602 | 4.120 |
| marriage | Marital status of the household: with a spouse = 1; without a spouse = 0 | 0.721 | 0.449 |
| health | Farmer's health status: very healthy = 1; healthy = 2; relatively healthy = 3; average = 4; unhealthy = 5 | 3.083 | 1.309 |
| lnworkincome | Non-agricultural income of farmers (CNY) | 5.375 | 4.995 |
| indo | Whether self-employed: yes = 1; no = 0 | 0.070 | 0.254 |
| fpop | Total household population | 4.021 | 2.033 |
| lnamach | Value of family-owned agricultural machinery (CNY) | 2.987 | 3.930 |
| lnfincome | Annual per capita net income of the family (CNY) | 9.184 | 0.864 |
| lnnetwork | Family annual favor expenditure (CNY) | 6.880 | 2.614 |
| distance | The distance from the village to the county | 53.51 | 41.61 |
| landform1 | Whether hills: yes = 1; no = 0 | 0.314 | 0.464 |
| landform2 | Whether mountains: yes = 1; no = 0 | 0.140 | 0.347 |
| landform3 | Whether plateaus: yes = 1; no = 0 | 0.084 | 0.277 |
| landform4 | Whether plains: yes-1; no = 0 | 0.403 | 0.491 |
| landform5 | Other terrains | 0.059 | 0.236 |
| internetp | The degree of importance of the internet as an information channel: important = 1; not important = 0 | 0.449 | 0.497 |

## 4. Empirical Analysis

### 4.1. Baseline Regression

Based on the model described above, the estimated results of the baseline regression are presented in Table 2. The first and third columns of Table 2 are estimates of the baseline equations for land rented-out and land rented-in, and the second and fourth columns add the interaction term (int_ser) of internet usage and agricultural socialization services. In both regressions, we directly report the average marginal effect.

**Table 2.** Basic regression to land-scale management.

| | Landout | | Landin | |
|---|---|---|---|---|
| Variable | Average Marginal Effect | Average Marginal Effect | Average Marginal Effect | Average Marginal Effect |
| internet | 0.0268 ** | 0.0447 *** | −0.00957 | −0.0127 |
| | (2.001) | (0.0167) | (0.0116) | (0.0179) |
| service | −0.114 *** | −0.100 *** | 0.0892 *** | 0.0875 *** |
| | (−11.06) | (0.0129) | (0.0100) | (0.0127) |
| int_ser | | −0.0366 * | | 0.00447 |
| | | (0.0206) | | (0.0196) |
| sex | −0.0118 | −0.0120 | −0.00634 | −0.00634 |
| | (−1.198) | (0.00987) | (0.00909) | (0.00909) |
| age | 0.00183 *** | 0.00188 *** | −0.00176 *** | −0.00177 *** |
| | (4.438) | (0.000413) | (0.000395) | (0.000396) |

| | | | | |
|---|---|---|---|---|
| edu | −0.00163 | −0.00173 | −0.00351 ** | −0.00351 ** |
| | (−1.000) | (0.00163) | (0.00145) | (0.00145) |
| marriage | −0.0465 *** | −0.0479 *** | 0.0374 *** | 0.0375 *** |
| | (−4.000) | (0.0116) | (0.0123) | (0.0123) |
| health | 0.00669 | 0.00662 | 0.00197 | 0.00198 |
| | (1.638) | (0.00408) | (0.00379) | (0.00379) |
| lnworkincome | 0.00427 *** | 0.00430 *** | −0.00140 | −0.00140 |
| | (3.825) | (0.00112) | (0.000994) | (0.000994) |
| indo | 0.0401 ** | 0.0403 ** | −0.00634 | −0.00628 |
| | (2.104) | (0.0191) | (0.0177) | (0.0177) |
| fpop | −0.00661 ** | −0.00659 ** | 0.00263 | 0.00261 |
| | (−2.289) | (0.00289) | (0.00254) | (0.00254) |
| lnamach | −0.00754 *** | −0.00751 *** | 0.00966 *** | 0.00965 *** |
| | (−5.415) | (0.00139) | (0.00112) | (0.00112) |
| lnfincome | 0.00771 | 0.00710 | 0.0121 ** | 0.0122 ** |
| | (1.236) | (0.00624) | (0.00600) | (0.00601) |
| lnnetwork | 0.00683 *** | 0.00672 *** | 0.00418 ** | 0.00418 ** |
| | (3.339) | (0.00204) | (0.00210) | (0.00210) |
| distance | −0.000364 *** | −0.000363 *** | −0.000125 | −0.000126 |
| | (−2.821) | (0.000129) | (0.000115) | (0.000115) |
| landform1 | −0.0462 ** | −0.0458 ** | 0.0316 | 0.0315 |
| | (−2.182) | (0.0211) | (0.0215) | (0.0215) |
| landform2 | −0.0664 *** | −0.0658 *** | 0.0176 | 0.0175 |
| | (−2.730) | (0.0243) | (0.0237) | (0.0237) |
| landform3 | −0.0368 | −0.0360 | 0.00133 | 0.00120 |
| | (−1.411) | (0.0260) | (0.0256) | (0.0256) |
| landform4 | 0.0241 | 0.0246 | 0.0288 | 0.0287 |
| | (1.189) | (0.0202) | (0.0210) | (0.0210) |
| landform5 | - | - | - | - |

N = 4982. *** $p < 0.01$, ** $p < 0.05$.

From the regression results in Table 2, for the land rented-out equation, the coefficient of internet usage is positive, and the average marginal effect is 4.47%, which indicates that farmers' access to agricultural information through the internet significantly promotes renting-out of the land. The coefficient of agricultural socialization service is negative, and the average marginal effect is −10%, which indicates that agricultural socialization service has an inhibitory effect on agricultural land-scale management. For the land rented-in equation, the coefficient of internet usage is not significant, but the coefficient of agricultural socialization service is positive at the 1% significance level. This is consistent with our assumptions one and two.

In the second and fourth columns, we introduce the interaction term for the internet and agricultural socialization services. Exploring the moderating effect of the internet, we found that the moderating effect only exists in the land rented-out equation and is significantly negative at the 10% level, indicating that, to a certain extent, the use of the internet will reduce the inhibitory effect of agricultural socialization services on land rented-out. At the same time, after the introduction of the interaction term, the impact of the internet increased by about 66.8%, and the inhibitory effect of agricultural socialization services decreased by about 12.3%. The possible reason is that being able to obtain agricultural socialization services reduces farmers' willingness to rented-out land, but the internet enables farmers to obtain more information about agricultural socialization services, which reduces farmers' transaction costs in agricultural production [12], so that they can

optimize their decisions based on new information and increase their willingness to rented-out land.

From the regression estimation results of the coefficients of other variables, in the household-level data, the influence of age on land transfer is significantly positive in the rented-out equation and negative in the rented-in equation, indicating that the opportunity cost for young farmers to engage in agriculture production activities is higher, and as they grow older, farmers are more inclined to rented-in land for agricultural production. The education factor (edu) is significantly negative in the land rented-in equation, indicating that the higher the education level, the higher the opportunity cost of engaging in agricultural production, and the lower the probability of willingness to rent-in. The marriage factor (marriage) is significantly negative in the land rented-out equation, and significantly positive in the land rented-in equation, which indicates that farmers without a spouse have a high probability of the rented-out, and farmers with a spouse have a high probability of the rented-in. Migrant labor income (lnworkincome) is significantly positive for the land rented-out and not significant for the land rented-in. The factor of self-employment (indo) significantly increases the probability of the land being rented-out but has no significant effect on rented-in. Gender (sex) and health (health) factors were not significant. In the family characteristics, the more the total family population (fpop), the more inhibited the probability of the land being rented-out, and the more increased the probability of the land being rented. The higher the family's agricultural machinery value (lnamach), the lower the probability of the land being rented-out, and the greater the probability of the land being rented. Household per capita income (lnfincome) has no significant effect on either the land rented-out or the land rented-in. The influence of the human expenditure (lnnetwork) factor is a significant positive effect. Among the village-level characteristic control variables, the closer the distance to the county town, the lower the probability of the land being rented-out, but the effect on the land rented-in is not significant. Among topographic factors, hills and mountains have a significant negative effect on the land rented-out but no significant effect on the land rented-in. The above conclusions are consistent with the research conclusions of scholars [12,51,52].

### 4.2. Heterogeneity Analysis

In this paper, the heterogeneity analysis of farmers in different income groups and different regions is carried out, and the results are shown in Tables 3 and 4. We found that the internet has a significant impact on high-income and middle-to-high-income farmers in the land rented-out equation. It indicates that high-income and middle-to-high-income rural residents have lower costs of using new technologies and can use the internet more effectively to obtain information, which also allows them to obtain more non-agricultural employment opportunities or join enterprises such as agricultural cooperatives to obtain higher income. However, the opportunity cost of engaging in agricultural production increases, which leads to an increase in the probability of the land being rented-out.

**Table 3.** Heterogeneity Analysis of Farmers Grouped by Income.

| Variable | Landout | | | | Landin | | | |
|---|---|---|---|---|---|---|---|---|
| | High | Middle to High | Low and Middle | Low | High | Middle to High | Low and Middle | Low |
| internet | 0.125 ** | 0.106 *** | 0.039 | −0.014 | 0.100 | −0.024 | −0.008 | −0.012 |
| | (2.163) | (0.0361) | (0.029) | (0.029) | (1.517) | (0.043) | (0.035) | (0.027) |
| service | −0.062 | −0.046 | −0.099 *** | −0.123 *** | 0.216 *** | 0.112 *** | 0.110 *** | 0.038 ** |
| | (−1.004) | (0.032) | (0.023) | (0.018) | (3.539) | (0.032) | (0.025) | (0.017) |
| Farmer characteristics | control | control | control | control | control | control | control | control |
| Family characteristics | control | control | control | control | control | control | control | control |
| Village characteristics | control | control | control | control | control | control | control | control |
| observations | 423 | 974 | 1582 | 2003 | 423 | 974 | 1582 | 2003 |

*** $p < 0.01$, ** $p < 0.05$.

**Table 4.** Heterogeneity Analysis of Farmers Grouped by Region.

| Variable | Landout | | | Landin | | |
|---|---|---|---|---|---|---|
| | Eastern | Central | Western | Eastern | Central | Western |
| internet | 0.067 ** | 0.089 ** | 0.021 | −0.003 | −0.094 ** | −0.002 |
| | (2.004) | (0.038) | (0.025) | (0.033) | (0.045) | (0.029) |
| service | −0.082 *** | −0.212 *** | −0.037 * | 0.087 *** | 0.127 *** | 0.067 *** |
| | (−3.166) | (0.029) | (0.019) | (0.023) | (0.030) | (0.021) |
| Farmer characteristics | control | control | control | control | control | control |
| Family characteristics | control | control | control | control | control | control |
| Village characteristics | control | control | control | control | control | control |
| observations | 1326 | 1257 | 1740 | 1326 | 1257 | 1740 |

*** $p < 0.01$, ** $p < 0.05$, * $p < 0.1$.

For agricultural socialization services, the impact on high-income and middle-to-high-income groups is not significant, but it has a significant negative impact on the decision making of low-to-middle-income and low-income groups, that is, low-income farmers are limited by their income levels and rely more on agricultural socialization service. If the level of agricultural socialization service is higher, the income of these farmers in agricultural production will be higher, and the distance between agricultural income and non-agricultural income will be leveled [1], thereby reducing the probability of the land being rented-out. For the land rented-in equation, the influence of the internet is not significant, and the agricultural socialization services have a significant positive impact on farmers' land-scale management decisions to reduce agricultural production costs.

According to the results in Table 4, we found that there is heterogeneity in the impact of the internet and agricultural socialization services on farmers' land-scale management decisions in different regions. For the land rented-out equation, the internet and agricultural socialization services have a significant impact on farmers' decision-making on land-scale management in the eastern and central regions, but have no significant impact on farmers in the western region. This shows that, in the eastern and central regions, the ability of farmers to use the internet to obtain information is higher than that in the western region. Agricultural socialization services have a significant negative impact on farmers' land lease decisions in different regions. However, the influence coefficients of different regions vary greatly, and the absolute value of the coefficient in the western region is significantly smaller than that in other regions. The reason may be that the promotion of agricultural socialization services in the western region is insufficient [38]. For the land rented-in equation, the effects of agricultural socialization services are all significantly positive. Regarding the impact of internet usage, only the central region has a significant negative impact on the land rented-in. This may be because, with the implementation of the strategy of the rise of central China, farmers in the central region have more opportunities to engage in non-agricultural operations. Using the internet, farmers can obtain more information, leading to a decreased willingness to rent-in.

*4.3. Robustness Analysis*

To test whether the regression coefficient results are robust and reliable, this paper further conducts a robustness test through sample adjustment and variables. Among the peasant households participating in land transfer, 43 peasant households both rented-out and rented-in. To verify the robustness of the estimated results, this paper removes these 43 peasant households from the sample and keeps all the core variables and control variables unchanged. Then, regression is re-run, and the results are shown in the first two columns of Table 5. The results are consistent with the basic model, indicating the robustness of the estimation results.

**Table 5.** Robustness check.

| Variable | Landout | Landin | Landout | Landin |
|---|---|---|---|---|
| internet | 0.044 *** | −0.016 | | |
| | (2.707) | (0.018) | | |
| internetp | | | 0.029 * | −0.024 |
| | | | (0.015) | (0.017) |
| service | −0.108 *** | 0.080 *** | −0.098 *** | 0.083 *** |
| | (−8.557) | (0.012) | (0.014) | (0.013) |
| Farmer characteristics | control | control | control | control |
| Family characteristics | control | control | control | control |
| Village characteristics | control | control | control | control |
| Observations | 4939 | 4939 | 4982 | 4982 |

*** $p < 0.01$, * $p < 0.1$.

To further verify the robustness of the results, we replaced the dummy variable of using the internet with the importance of farmers on the internet as an information channel, with all other variables unchanged. The results are listed in the third and fourth columns of Table 5, and the estimated results are consistent with the basic model, which once again confirms the robustness of the results.

## 5. Conclusions

This paper analyzed whether the internet and agricultural socialization services affect farmers' decision making on land-scale management using micro-survey data from the China Family Panel Studies (CFPS) of Peking University in 2018. The research results show that: first, the use of the internet significantly promoted farmers' land rented-out behavior but has no significant impact on the land rented-in behavior. This result is consistent with Hypothesis 1 of this paper and in line with previous studies. Our result suggests that the introduction of new technologies such as the internet makes information access easier, reduces the cost of information asymmetry, and affects farmers' decisions on land-scale management.

Secondly, agricultural socialization services may inhibit farmers' land rented-out decisions, and stimulate farmers' land rented-in behaviors. This result is consistent with Hypothesis 2 of this paper. Since the sunk cost of purchasing agricultural machinery for individuals is too high, this highlights the huge cost advantage of agricultural socialization services. Therefore, we suggest that the availability of agricultural socialization services may reduce the production cost of farmers and improve production efficiency. Meanwhile, our findings found that when the land is rented-out, the internet has a moderating effect on agricultural socialization services.

Thirdly, the heterogeneity analysis shows that the impact of the internet and agricultural socialization services on farmers' land-scale management decisions would have different effects due to different income levels and different regions. These results are consistent with Hypothesis 3 of this paper. We found that the internet has a significant impact on high-income and middle-to-high-income farmers in the land rented-out. This indicates

that these farmers have lower costs of using new technologies, and more effectively obtain information, which also allows them to obtain more non-agricultural employment opportunities and higher income. For agricultural socialization services, the impact on high-income and middle-to-high-income groups is not significant, but it has a significant negative impact on the decision making of low-income groups, that is, low-income farmers may rely more on agricultural socialization services. Meanwhile, the results of the analysis of regional heterogeneity show that, for the land rented-out, the internet and agricultural socialization services have a significant impact on farmers' decision making on land-scale management in the eastern and central regions but have no significant impact on farmers in the western region. For the land rented-in, the effects of agricultural socialization services are all significantly positive. Regarding the impact of internet usage, only the central region has a significant negative impact on the land rented-in.

In this study, we analyzed how the introduction of new technologies and new systems affects farmers' decision making on land-scale management. Our analysis concludes that making internet and agricultural socialization services benefit all farmers, helping to more effectively improve the efficiency of rural land use and promote the optimal allocation of rural resources. Although in line with previous studies' results, our analysis conducted a more detailed analysis of heterogeneity in different regions and different income levels. Our outcomes would more reflect recent actual conditions if the study sample period is extended for evaluation. Future researchers should consider using the latest data to study the determinants of new technologies and institutional arrangements on farmers' land-scale management, and may also extend the research to the evaluation of institutional performance.

With the continuous reduction in the cost of using the internet, and its popularization in rural areas, the internet may provide more farmers with the information needed for agricultural production. The development of agricultural socialization services also allows farmers to share the dividends of agricultural technological progress and promote improvement in rural production efficiency. Rural land is the most important means of agricultural production, and many studies have shown that the effective way to improve land-use efficiency is moderatelandscale management. The modern agricultural production process has gradually separated from labor-intensive production methods, and new technologies have been continuously combined with traditional agriculture, resulting in emerging production methods such as "internet + agriculture". Various types of family farms and rural professional cooperatives are constantly emerging, which allows more farmers to transfer their land to engage in non-agricultural work or work for agricultural enterprises.

At the same time, we should also pay attention to the heterogeneity of the internet for different income groups. According to the research results, farmers in areas with higher levels of economic development have better opportunities to obtain off-farm jobs, so they are more inclined to rent-out the land. The eastern region of China is subject to an increase in production costs, especially an increase in labor prices, which has prompted a large number of labor-intensive industries to move westward. Agricultural job opportunities are required, so that farmers in the central region can better maintain the balance between non-agricultural work and agricultural production. Due to the huge digital divide in internet technology, farmers will have significant differences in their ability to access and use the internet due to their geographic location, education level, internet penetration, and information infrastructure. Middle-to-high-income and high-income farmers can use the internet more effectively, but low-income farmers cannot. Therefore, it is the focus of the next policy on how to make the new technologies benefit all farmers in China.

**Author Contributions:** Conceptualization, X.L., and H.L.; methodology, X.L; formal analysis, X.L., and H.L.; data curation, X.L.; writing—original draft preparation, X.L.; writing—review and editing, H.L. All authors have read and agreed to the published version of the manuscript.

**Funding:** This work was supported by the Pai Chai University research grant in 2022.

**Institutional Review Board Statement:** Not applicable.

**Informed Consent Statement:** Not applicable.

**Data Availability Statement:** The CFPS data for this analysis come from Peking University Open Research Data: https://opendata.pku.edu.cn/dataverse/CFPS?language=en [accessed on 25 February 2022].

**Conflicts of Interest:** The authors declare no conflict of interest.

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
