# Peer review of "An Analysis on the Determining Factors of Farmers’ Land-Scale Management: Empirical Analysis Based on the Micro-Perspective of Farmers in China"

_land, doi:10.3390/land11081278_

Round 1

Reviewer 1 Report

The subject of the article is interesting and it is, somehow, linked to the objectives of the journal). However, there are a number of issues that have to be reconsidered.

For better visibility on databases, the authors are asked not to repeat among keywords the words/concepts included in the title of the article.

The introduction is too long. This should not be a review of research, but an introduction to the topic of your study Propose only to focus on the main subject. Alos, the presentation of the structure of the paper is missing.

 In the methodology, there is necessary to explain in more detail how the sample was selected. Also, Figure 2 is useless. 

Discussions are quite simplest and not enough elaborated,

the Conclusions. Limits, possibilities of study follow-up, and recommendations for scholars, companies, and government must be more precise. 

Author Response

COVER LETTER FOR REVIEWER 1

Dear reviewer,

We are grateful to the reviewers for your time and constructive comments on our manuscript. We have implemented your comments and suggestions and wish to submit a revised version of the manuscript for further consideration in the journal. Changes in the initial version of the manuscript are marked up using the “Track Changes” function in the revised version. In order to reflect the suggestions of the reviewers, the revised manuscript is a bit messy, we hope for understand. Below, we also provide a point-by-point response explaining how we have addressed each of the reviewers’ comments. We look forward to the outcome of your assessment

Yours sincerely,

On behalf of the co-author

Lee, Hyukku

Reviewer 1:

Comments to the Author:

The subject of the article is interesting and it is, somehow, linked to the objectives of the journal. However, there are a number of issues that have to be reconsidered.

Our answer: Thanks for your appreciation.

Comment 1. For better visibility on databases, the authors are asked not to repeat among keywords the words/concepts included in the title of the article.

Our answer: Thanks for your important advice, we adjusted the title (“An Analysis on the Determining Factors of Farmers' Land Scale Management: Empirical analysis based on the micro-perspective of farmers in China”) and keywords (land scale management; internet; agricultural socialization services; probit model; heterogeneity, rural land, farmer, China). Please see Line 2-5 of the main body of the revised manuscript.

Comment 2. The introduction is too long. This should not be a review of research, but an introduction to the topic of your study Propose only to focus on the main subject. Also, the presentation of the structure of the paper is missing.

Our answer: Thanks for your raising this point, as you mentioned, the introduction is narrowed down to the core. Please see Chapter1 introduction of the main body of the revised manuscript. The structure of the paper is presented in the last part of Chapter 1.

Comment 3. In the methodology, there is necessary to explain in more detail how the sample was selected. Also, Figure 2 is useless.

Our answer: Thanks for pointing out this. As requested, we have added the method explained in Section3. Please see Line 430-441 of the main body of the revised manuscript. However, there is no figure in our manuscript.

Comment 4. Discussions are quite simplest and not enough elaborated.

Our answer: Thanks for pointing out this. As requested, we have added some discussion in Section 5. Please see Line 646-674 of the main body of the revised manuscript.

Comment 5. the Conclusions. Limits, possibilities of study follow-up, and recommendations for scholars, companies, and government must be more precise.

Our answer: Thanks for pointing out this. We revised the conclusion. As you requested, the limitations of the study and implications for follow-up studies were supplemented. Please see Line 675-711 of the main body of the revised manuscript.

Reviewer 2 Report

1. Line 122-125: These Roman numerals should be converted into Arabic numerals.

2. To keep the context consistent, the 2.1 subheadings 2.1 and 2.2 should be adjusted as follows: 2.1 Internet and Land Scale Management, 2.2 Agricultural Socialization Service and Land Scale Management.

3. Are land outflow and land inflow generally accepted terms?

4. Why not take logarithms of variables related to money? Such as non-agricultural income of farmers, Annual per capita net income of the family ?

5. Generally speaking, the higher the per capita income of a family, the higher its non-agricultural income. However, the author takes these two variables as control variables at the same time, which may lead to collinearity. Actually, the authors can use the can use the non-agricultural income ratio to weaken the collinearity problem.

6. The author considers the influence of landform, but it is difficult to understand why landform is treated as an ordered variable. Have you compared the land transfer-in ratio and land transfer-out ratio between different landforms?

7. Please add collinearity test results.

8. It is recommended to re-read the text to eliminate small typos.

9. Introduce a discussion section whereby you can use state-of-the art knowledge to revamp it for scientific community especially to generate new knowledge.

Author Response

COVER LETTER FOR REVIEWER 2

Dear reviewer,

We are grateful to the reviewers for your time and constructive comments on our manuscript. We have implemented your comments and suggestions and wish to submit a revised version of the manuscript for further consideration in the journal. Changes in the initial version of the manuscript are marked up using the “Track Changes” function in the revised version. In order to reflect the suggestions of the reviewers, the revised manuscript is a bit messy, we hope for understand. Below, we also provide a point-by-point response explaining how we have addressed each of the reviewers’ comments. Please see the attachment. We look forward to the outcome of your assessment

Yours sincerely,

On behalf of the co-author

Lee, Hyukku

Reviewer 3 Report

Please see 22 comments in the attached word file.

Author Response

COVER LETTER FOR REVIEWER 3

Dear reviewer,

We are grateful to the reviewers for your time and constructive comments on our manuscript. We have implemented your comments and suggestions and wish to submit a revised version of the manuscript for further consideration in the journal. Changes in the initial version of the manuscript are marked up using the “Track Changes” function in the revised version. In order to reflect the suggestions of the reviewers, the revised manuscript is a bit messy, we hope for understand. Below, we also provide a point-by-point response explaining how we have addressed each of the reviewers’ comments. Please see the attachment. We look forward to the outcome of your assessment

Yours sincerely,

On behalf of the co-author

Lee, Hyukku

Reviewer 4 Report

An important and current research problem was taken up in the manuscript submitted for review. The developed issue is important both from a scientific and practical point of view. Research methods and tools (including statistical) were properly selected and applied. Too superficial treatment of the theoretical background in relation to the theory of ownership rights and the theory of agencies (including information asymmetry) raises a certain insufficiency. In the opinion of the reviewer, it would be worth developing this thread.

The following scientific publications may be helpful:

Spence, M., Zeckhauser, R., 1971. Insurance, Information and Individual Action. American Economic Review (May, 1971). Papers and Proceedings of the Eighty-Third Annual Meeting of the American Economic Association Published by: American Economic Association, Vol. 61, No 2, 380-387.

Ross, S. A., 1973. The Economic Theory of Agency: The Principal’s Problem. American Economic Review, 63 (2), 134-139. https://www.jstor.org/stable/1817064

Stiglitz, J. E., 1974. Incentives and risk sharing in sharecropping, Review of Economic Studies, 41 (April), 219–255. https://www.jstor.org/stable/pdf/2296714.pdf

Bromley D. W. 1991. Environment and the Economy. Property Rights and Public Policy. Basil Blackwell.

Benson B. L. 1994. Emerging from the Hobbesian Jungle: Might Takes and Makes Right. Constitutional Political Economy, Vol 5, No 2.

Beckmann V. 2000. Transaktionskosten und institutionelle Wahl in der Landwirtschaft: Zwischen Markt, Hierarchie und Kooperation. Edition Sigma Rainer Bohn, Verlag, Berlin.

Marks-Bielska R. 2021. Conditions underlying agricultural land lease in Poland, in the context of the agency theory. Land Use Policy, ELSEVIER, 102(2021)105251 https://doi.org/10.1016/j.landusepol.2020.105251

Author Response

COVER LETTER FOR REVIEWER 4

Dear reviewer,

We are grateful to the reviewers for your time and constructive comments on our manuscript. We have implemented your comments and suggestions and wish to submit a revised version of the manuscript for further consideration in the journal. Changes in the initial version of the manuscript are marked up using the “Track Changes” function in the revised version. In order to reflect the suggestions of the reviewers, the revised manuscript is a bit messy, we hope for understand. Below, we also provide a point-by-point response explaining how we have addressed each of the reviewers’ comments. We look forward to the outcome of your assessment

Yours sincerely,

On behalf of the co-author

Lee, Hyukku

Reviewer 4:

An important and current research problem was taken up in the manuscript submitted for review. The developed issue is important both from a scientific and practical point of view. Research methods and tools (including statistical) were properly selected and applied.

Our answer: Thanks for your appreciation.

Too superficial treatment of the theoretical background in relation to the theory of ownership rights and the theory of agencies (including information asymmetry) raises a certain insufficiency. In the opinion of the reviewer, it would be worth developing this thread.

Our answer: We fully agree with the points raised by the reviewer. We slightly supplement the literature on agency theory and property rights theory proposed by you. According to the reviewer's suggestion, we will expand the research direction for follow-up research.

The following scientific publications may be helpful:

Spence, M., Zeckhauser, R., 1971. Insurance, Information and Individual Action. American Economic Review (May, 1971). Papers and Proceedings of the Eighty-Third Annual Meeting of the American Economic Association Published by: American Economic Association, Vol. 61, No 2, 380-387. 

Ross, S. A., 1973. The Economic Theory of Agency: The Principal’s Problem. American Economic Review, 63 (2), 134-139. https://www.jstor.org/stable/1817064 

Stiglitz, J. E., 1974. Incentives and risk sharing in sharecropping, Review of Economic Studies, 41 (April), 219–255. https://www.jstor.org/stable/pdf/2296714.pdf 

Bromley D. W. 1991. Environment and the Economy. Property Rights and Public Policy. Basil Blackwell. 

Benson B. L. 1994. Emerging from the Hobbesian Jungle: Might Takes and Makes Right. Constitutional Political Economy, Vol 5, No 2. 

Beckmann V. 2000. Transaktionskosten und institutionelle Wahl in der Landwirtschaft: Zwischen Markt, Hierarchie und Kooperation. Edition Sigma Rainer Bohn, Verlag, Berlin. 

Marks-Bielska R. 2021. Conditions underlying agricultural land lease in Poland, in the context of the agency theory. Land Use Policy, ELSEVIER, 102(2021)105251 https://doi.org/10.1016/j.landusepol.2020.105251

Our answer: We are very grateful for the kinds of literature provided by the reviewer, which enriched our research.

Round 2

Reviewer 1 Report

The authors succeeded in answering my concerns. The article could be published.

Author Response

Dear, Reviewer 1

We are grateful to the reviewer 1 for your time and constructive comments on our manuscript in the revision.

Comment to the Author: The authors succeeded in answering my concerns. The article could be published.

Our answer: Thanks for your appreciation.

Reviewer 2 Report

Did a good job, but the authors did not answer why landform types is treated as ordered variable!

Author Response

Dear, Reviewer 2,

We are grateful to the reviewer 2 for your time and constructive comments on our manuscript in the revision.

Comment to the Author: Did a good job, but the authors did not answer why landform types is treated as ordered variable!

Our answer: We are very sorry that we forgot to answer an important question in the first revision. First of all, thanks again to the reviewer for raising this critical question. Our original intention was to control the terrain factor, but due to our negligence in our work, we mistakenly included the terrain factor in the regression as an ordinal variable, which is a very serious mistake. We actively corrected this error and controlled for terrain factors in the form of dummy variables.

At the same time, because the sample size of some terrains in the sample is too small, we have merged them. Finally, we controlled five groups of terrain control variables, namely: landform1 (whether hills or not), landform2 (whether mountain or not), landform3 (whether plateaus or not), landform4 (whether plains or not), and landform5 (other terrains).

For a detailed statistical description, see table 1 in the revised manuscript. And, we also revised all the regression results (see table 2-5) and texts (see Line 441, 504, 507,518,519, 527, 551, 555-556, 609-613, 629, 631).

After controlling for the terrain dummy variables, there was no change in significance or sign of key variables. The main conclusions of the article are still robust.